# Scoping review on regulation, implementation and postmarket surveillance of medical devices

**Mathias Damkjær**[1,2]*, **Mia Elkjær**[1,2], **Asbjørn Hróbjartsson**[1,2], **Jeppe B. Schroll**[1,2,3]

**1** Cochrane Denmark & Centre for Evidence-Based Medicine Odense, Department of Clinical Research, University of Southern Denmark, Odense, Denmark **2** Open Patient data Explorative Network (OPEN), Odense University Hospital, Denmark, **3** Gynecology & Obstetric Department, Herlev and Gentofte University Hospital, Denmark

* mwdamkjaer@health.sdu.dk

## Abstract

### Background

Regulation and postmarket surveillance of medical devices have been criticized for being too lenient as compared to drug regulation and postmarket surveillance. Little is known about the factors that determine which medical devices are chosen for implementation among similar medical devices.

### Objective

Our aim was to systematically identify and characterize empirical studies on medical device regulation, implementation and postmarket surveillance, and to describe the recurring themes and trends in the studies.

### Methods

The scoping review was preregistered, with the protocol publicly available (https://osf.io/mx36f). We followed the JBI guidelines for scoping reviews and reported the review in accordance with PRISMA-ScR guidelines. Last searches were done in MEDLINE and Embase through Ovid on 8th of February 2024. We included primary studies with empirical data, and we excluded any secondary studies such as editorials, opinion papers or systematic reviews using bibliographic databases as the primary data source. We were interested in studies that examined medical devices approved by the U.S. Food and Drug administration (FDA) and European Union (EU), and any studies on the decision-making process regarding medical device implementation.

We described study characteristics and mapped them graphically. Recurring themes were presented in a table. Furthermore, we reported conclusions from identified essential studies and provided a summary of the main results. Graphs and descriptive statistics were done in R version 4.3.2, package ggplot2.

**Data availability statement:** All relevant data are within the paper and its Supporting information files, and at the Open Science Framework fileserver: https://osf.io/nj3be/files/osfstorage.

**Funding:** The author(s) received no specific funding for this work.

**Competing interests:** The authors have declared that no competing interests exist.

## Results

We screened 3862 titles/abstracts, after which 368 records were assessed in full-text, yielding 139 studies included in the review. Out of these, 68 studies (49%) examined approval, 40 studies (29%) examined postmarket surveillance, 17 studies (12%) implementation and 14 studies (10%) both approval and postmarket surveillance. The studies were published between 2003–2024 and consisted of 77 cross-sectional studies (55%), 35 cohort studies (25%), 20 qualitative studies (14%) and seven mixed-methods studies (5%). As data source, 90 studies (65%) used FDA, 25 studies (18%) other data sources and 24 studies (17%) interviewees through semi-structured interviews. Nine out of the 139 studies investigated regulatory approval within the EU. Predominantly, the studies reported that the available clinical evidence for medical device approval was considered inadequate, making it difficult for stakeholders to evaluate the suitability of a medical device for implementation.

## Conclusions

Studies on medical devices are mainly conducted using FDA device databases, since restricted access to publicly available data has hindered research within the EU. Research on how and why specific medical devices are chosen and adopted into clinical practice is limited. We suggest that evidence on medical device efficacy and harms should be strengthened through higher demands from regulatory agencies and improvement of accessible registries.

## Introduction

Medical devices are fundamental to medicine, and the basis for a growing billion-dollar industry [1,2]. The World Health Organization estimates that there are more than two million medical devices on the world market and they range from pacemakers, artificial intelligence software to simple instruments such as scalpels [3]. The regulation of medical devices faces several challenges, including inconsistent approval standards across different devices and countries, as well as difficulties in monitoring device harms and efficacy once they are in use. Medical devices are regulated by agencies such as the U.S. Food and Drug Administration (FDA) [4] in the United States of America (USA) and by the European Union (EU) medical device regulation with member state level involvement such as national competent authorities [5].

Manufacturer applications for market approval of medical devices are submitted to regulatory authorities such as the FDA and private notified bodies within Europe. Private notified bodies are organisations or private companies designated by an EU Member Country to assess conformity of medical devices and give a conformité européenne certification. Conformité européenne certification signify that the medical devices have met applicable European safety, health and environmental protection requirements. Medical devices are classified by both the FDA and the EU into three risk classes I-III, which are further subdivided [6,7]. Class III is presumed to have the

highest risk for patients and counts for instance cardiovascular implants and require stricter evaluation for approval than class I (low risk) medical devices such as a bandage.

Premarket approval (PMA) [8] is the most rigorous pathway for market approval in USA and requires clinical data. It is the FDA process of evaluating the efficacy and harms of new class III medical devices. Modifications to a PMA approved device that can affect efficacy and harms requires a PMA supplement for review and approval by the FDA [9]. However, in the USA most medical devices are cleared by FDA through the less strict 510(k) regulation that states that if the medical device can claim substantial equivalence to a previous cleared device (after 1976), then it can avoid the more rigorous PMA process that novel class III medical devices must undertake. Substantial equivalence requires that the device has the same intended use and technological characteristics as the predicate device that is the medical device used for benchmarking. Usually, 510(k) clearance requires no clinical data, is faster, cheaper and has limited postmarket surveillance requirements compared to the PMA pathway. However, the FDA may sometimes demand biological and clinical performance data to demonstrate this claim [10]. Previously, some medical devices such as vaginal meshes and hip prostheses that were approved by FDA and/or in the EU eventually turned out to have severe adverse effects [11,12]. Among high-risk devices approved between 2008 and 2017 through FDA PMA approval, 27% were recalled as of December 31, 2019 [13]. A more stringent and transparent premarket and postmarket evaluation was called upon by top medical journals such as the Lancet and the British Medical Journal [14,15].

Regulatory institutions such as the European Commision promised more stringent and transparent medical device regulation with a regulation from 2017 [16]. The EU medical device regulation dictate for instance that from May 2024 class III devices require clinical data, a unique identification number, and a public summary of safety and clinical performance to be reported in a new European Database on Medical Devices (EUDAMED) [17], but the fully functional online database is delayed as of June 2024. Furthermore, all previously approved medical devices need to meet new legal requirements. Unlike in the EU, the FDA have had public available databases containing information since the early 2000s. For example, reporting of medical device adverse events through the Manufacturer and User Facility Device Experience (MAUDE) database [18].

We did not identify prior reviews that created an overview of the conducted research on medical devices across the selected key lifecycle domains – approval, implementation and postmarket surveillance – likely because the studies are very heterogeneous. Additionally, we hypothesized that studies investigating why one medical device is adopted over a similar alternative are limited.. Our aim was to systematically map and characterize empirical studies on regulation, postmarket surveillance and implementation of medical devices in a scoping review given the heterogeneity of studies. We sought to discuss strengths and limitations of these domains, based on a condensation of the predominant conclusions from included studies, while also identifying areas that could be improved, and where more research is needed.

## Methods

The study was preregistered, and the protocol was uploaded to Open Science Framework (https://osf.io/mx36f). The scoping review was conducted following the JBI Methods Manual for Scoping Reviews (JBI manual) [19–21] and reported according to PRISMA-ScR [22] for scoping reviews.

### Inclusion criteria

Inclusion criteria for the scoping review were based on the Population, Concept and Context (PCC)-framework (see S1 File Supporting information for details). We included studies that examined medical devices in relation to regulation, implementation, and postmarket surveillance. The regulation and postmarket surveillance domains were limited to FDA/ EU approved medical devices. Results from studies had to be generalizable, e.g., be relevant and applicable for a broader range of situations.

We included any empirical (primary) studies such as cohort, case-reports, cross-sectional or qualitative studies.

We excluded secondary papers, defined as publications that do not present primary empirical research, such as editorials, systematic reviews based on bibliographic databases, editorial letters, and opinion papers. Additionally, we excluded papers that did not fit the predefined PCC framework.

## Changes to protocol

We identified more studies than expected and chose to filter out studies on diagnostics or software assisted medical devices, e.g., artificial intelligence solutions, and medical devices not used for treatment purposes. Protocol amendments were prospectively documented at https://osf.io/mx36f.

## Search strategy and information sources

The search strategy was developed using an iterative process due to the complexity and heterogeneity of medical device research. Different key words were piloted through PubMed and later Ovid. Some of the considered key studies (see S1 File Supporting information) were screened for keywords within each concept in the PCC-framework, and the search was piloted ensuring that the search found preselected key studies. Search filters for observational studies, clinical trials and qualitative studies were found from Ovid tools and resources portal [23] and The InterTASC Information Specialists' Sub-Group Search Filter Resource [24]. An information specialist developed a supplementary search to retrieve additional studies that the main search might have missed.

Last full searches were conducted in the electronical bibliographic databases MEDLINE (All) and EMBASE (Classic+Embase) through Ovid on 8th of February 2024. A secondary search was conducted in Google scholar using free keywords with no restriction to language and sorted by relevance. Moreover, reference lists and forwards citations from included studies was screened for further studies through Web of Science. The database search strategy can be found in the S1 File Supporting information.

## Study selection and data charting

All records from the final search were exported to Covidence [25]. One author (MWD) screened records by title/abstract and excluded all records obviously not relevant. All remaining records were then abstract/full text reviewed for inclusion by two authors (MWD/JBS) independently. Disagreements on study inclusions were solved through discussion.

We predefined the following data charting items used for mapping in our protocol: author, title, regulatory authority (FDA/EU), country, publication year, study design, device categories (specific device, one category or two or more categories), risk class (class 1–3), data sources, data sampling years, adherence to a reporting guideline (yes/no), analysis adjusted for confounding (yes/no), study objective, funding (industry, non-industry or unclear), lifecycle domain (regulation, implementation or postmarket surveillance), DOI-link, conclusion and other information.

Study characteristics and information from the included studies was extracted by two independent authors (MWD/ME) using a predefined list of data items in Microsoft Access/Excel.

## Synthesis of the results

We mapped studies according to the protocol in three medical device lifecycle domains: regulation, implementation and postmarket surveillance. Study characteristics were descriptively presented in tables and graphically mapped. Recurring themes were mapped and presented in a table. We reviewed the results and conclusions from all the included studies, and reported the main conclusions that were most consistent across studies, while presenting and referencing essential studies. Finally, we created a brief and condensed summary of the main results. Descriptive statistics and plots were computed in R version 4.3.2 [26], packages ggplot2 and tidyverse.

## Results

The search identified 4,904 records. After de-duplication, 3,862 title/abstracts were screened and 368 full-text studies were assessed, out of which 141 met eligibility criteria, and 139 were included in the final analysis. We excluded 227 studies and the main reason for exclusion was because studies did not fit the defined Population, Concept and Context or was considered not generalizable (Fig 1). For a full list of see excluded studies and reasons for exclusion see the table in S2 File Excluded studies.

The 139 included studies were published between 2003–2024, out of which 68 studies (49%) were on approval, 40 studies (29%) on postmarket surveillance, 17 studies (12%) on implementation, 14 studies (10%) on both approval and postmarket surveillance. The studies consisted of 77 cross-sectional studies (55%), 35 cohort studies (25%), 20 qualitative studies (14%) and seven mixed-methods studies (5%) (Fig 2).

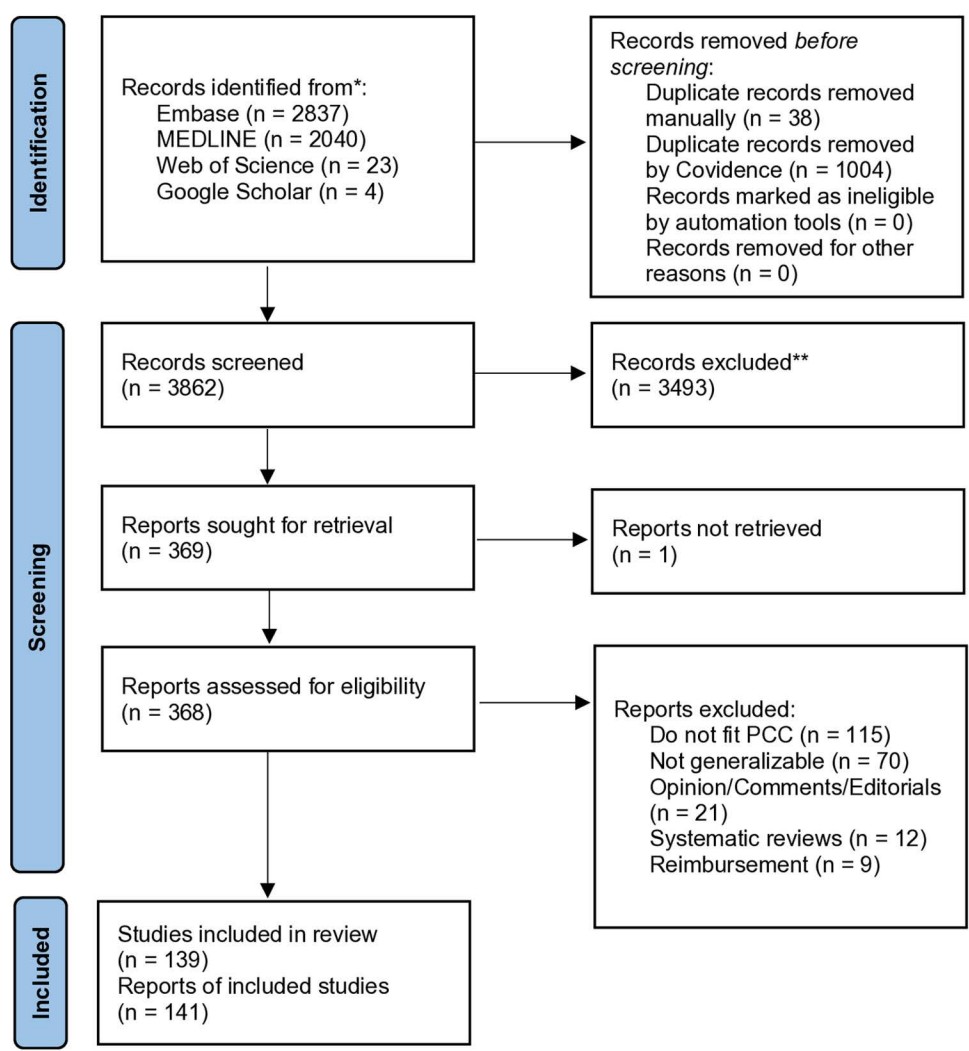

**Fig 1. PRISMA flow diagram.** PCC = Population, Concepts and Context.

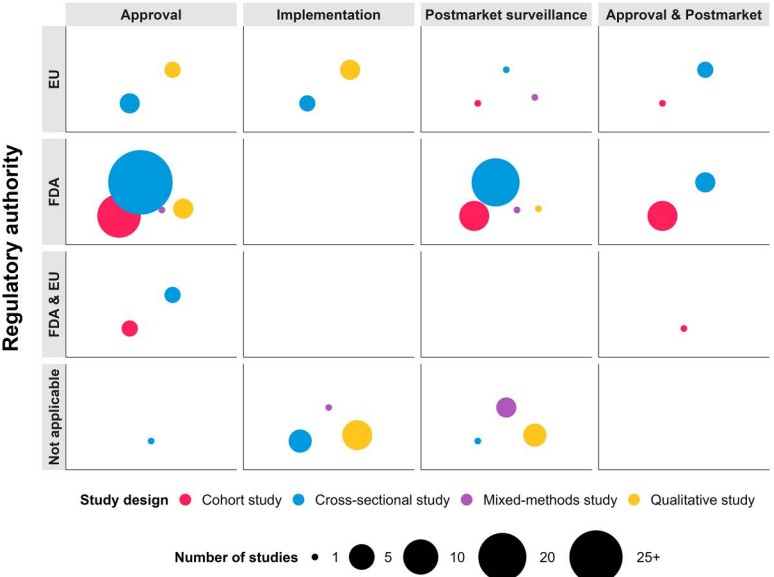

**Lifecycle domains of medical devices**

Study design ● Cohort study ● Cross-sectional study ● Mixed-methods study ● Qualitative study

Number of studies ● 1 ● 5 ● 10 ● 20 ● 25+

**Fig 2. Evidence-Map of identified studies mapped according to lifecycle domain and regulatory authority.**

As data source, 28% used the FDA PMA, 15% used semi-structured interviews or surveys, 13% used FDA 510(k) and 13% used Manufacturer and User Facility Device Experience (MAUDE), 12% used other FDA sources, 9% used the FDA recall database and 8% used other data sources, e.g., ethics committee applications or registries (Fig 3).

The medical device category investigated the most was cardiovascular devices (21 studies; 19%) followed by orthopedic devices (9 studies; 8%). We identified 16 studies (12%) that used a reporting guideline, and 3 studies (2%) were industry funded (Table 1).

The included studies were mapped according to themes (Table 2), and a full list of the included studies according to emerged themes is available in the S1 File Supporting information.

In the following sections selected highlights from the collected articles are reported according to each lifecycle domain.

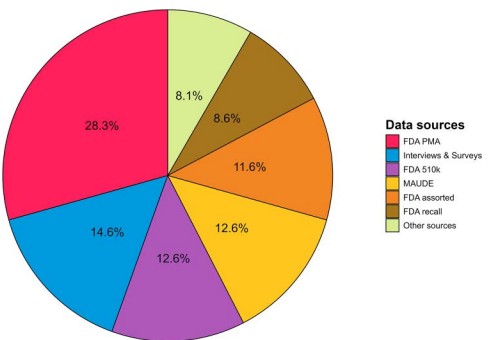

Data sources
- FDA PMA
- Interviews & Surveys
- FDA 510k
- MAUDE
- FDA assorted
- FDA recall
- Other sources

**Fig 3. Piechart of data sources used in the studies.**

**Table 1. Characteristics of included studies.**

| Study characteristics | Lifecycle domain | | | | |
| | Approval | Implementation | Postmarket | Approval & postmarket | Total (N = 139) |
|---|---|---|---|---|---|
| | Number of studies (%) | | | | |
| **Regulatory authority** | | | | | |
| FDA | 58 (85.3%) | 0 (0%) | 29 (72.5%) | 10 (71.4%) | 97 (69.8%) |
| EU | 5 (7.4%) | 5 (29.4%) | 3 (7.5%) | 3 (21.4%) | 16 (11.5%) |
| FDA & EU | 4 (5.9%) | 0 (0%) | 0 (0%) | 1 (7.1%) | 5 (3.6%) |
| Not applicable | 1 (1.5%) | 12 (70.6%) | 8 (20%) | 0 (0%) | 21 (15.1%) |
| Total | 68 (100%) | 17 (100%) | 40 (100%) | 14 (100%) | 139 (100%) |
| **Study design** | | | | | |
| Cross-sectional | 44 (64.7%) | 6 (35.3%) | 22 (55%) | 5 (35.7%) | 77 (55.4%) |
| Cohort | 18 (26.5%) | 0 (0%) | 8 (20%) | 9 (64.3%) | 35 (25.2%) |
| Qualitative | 5 (7.4%) | 1 (5.9%) | 5 (12.5%) | 0 (0%) | 11 (7.9%) |
| Mixed-methods | 1 (1.5%) | 10 (58.8%) | 5 (12.5%) | 0 (0%) | 16 (11.5%) |
| Total | 68 (100%) | 17 (100%) | 40 (100%) | 14 (100%) | 139 (100%) |
| **Adherence to reporting guideline** | | | | | |
| Yes | 4 (5.9%) | 3 (17.6%) | 6 (15%) | 3 (21.4%) | 15 (10.8%) |
| No | 64 (94.1%) | 14 (82.4%) | 34 (85%) | 11 (78.6%) | 124 (89.2%) |
| Total | 68 (100%) | 17 (100%) | 40 (100%) | 14 (100%) | 139 (100%) |
| **Risk class device** | | | | | |
| Class 2 | 5 (7.4%) | 0 (0%) | 3 (7.5%) | 1 (7.1%) | 9 (6.5%) |
| Class 2 and 3 | 8 (11.8%) | 1 (5.9%) | 15 (37.5%) | 6 (42.9%) | 30 (21.6%) |
| Class 3 | 40 (58.8%) | 4 (23.5%) | 11 (27.5%) | 6 (42.9%) | 61 (43.9%) |
| Mixed/Not applicable/Not reported | 15 (22%) | 12 (70.6%) | 11 (27.5%) | 1 (7.1%) | 39 (28.1%) |
| Total | 68 (100%) | 17 (100%) | 40 (100%) | 14 (100%) | 139 (100%) |
| **Industry funded** | | | | | |
| Yes | 0 (0%) | 0 (0%) | 3 (7.5%) | 0 (0%) | 3 (2.2%) |
| No | 40 (58.8%) | 8 (47.1%) | 18 (45%) | 10 (71.4%) | 76 (54.7%) |
| Not reported | 28 (41.2%) | 9 (52.9%) | 19 (47.5%) | 4 (28.6%) | 60 (43.2%) |
| Total | 68 (100%) | 17 (100%) | 40 (100%) | 14 (100%) | 139 (100%) |

FDA = Food and Drug Administration; EU = European Union.

### Regulatory approval

A total of 68 studies reported on medical devices in relation to approval. In the following sections 28 studies across five themes are presented.

### FDA premarket approval pathway

Studies [27–31] concluded that premarket approval studies on high-risk cardiovascular devices lacked adequate strength, were poorly reported, may be prone to bias, did not use an active control group, and frequently used surrogate endpoints with a short follow-up period. Also, it was reported that between 20–51% of premarket approval studies for high-risk cardiovascular devices remained unpublished [32,33]. Furthermore, it was found hard to determine when evidence for a medical device was considered sufficient among stakeholders with experience on medical device certification [34,35]. In the qualitative study [34] a stakeholder said: "*For example, when asked: 'when is clinical evidence sufficient enough?',*

**Table 2. Overview of recurring patterns from the empirical studies.**

| Thematic theme | Frequency (n) | % |
|---|---|---|
| Qualitative studies on various thematic categories | 22 | 15.8 |
| Approval through the PMA pathway | 21 | 15.1 |
| Postmarket surveillance miscellaneous | 18 | 12.9 |
| Approval or clearance miscellaneous | 16 | 11.5 |
| FDA 510k versus PMA pathway | 13 | 9.4 |
| Generalizability of PMA medical devices | 12 | 8.6 |
| PMA approval amendments | 8 | 5.8 |
| Implementation miscellaneous | 7 | 5.0 |
| FDA versus EU | 6 | 4.3 |
| The MAUDE database solely | 5 | 3.6 |
| Premarket versus postmarket evidence | 5 | 3.6 |
| Predicate device used for clearance | 4 | 2.9 |
| Drug versus device regulation | 2 | 1.4 |

one consultant replied, 'no one knows… there's no agreement on what clinical evidence is under the (EU medical device regulation) MDR'".

## Generalizability of medical device trials

We identified 12 studies that all concluded that premarket approval studies did not enroll a diverse study population, e.g., sex-specific, pediatric and non-white population estimates, causing a concern for lack of generalizability of approved medical devices.

## Mandated postmarket approval studies

A trend towards less strict clinical data at market approval and increased FDA mandated postmarket approval studies was reported in a study on endovascular devices [36]. It was further reported [37,38], that one out of three of the FDA mandated postmarket approval studies were observed to have been completed and published within eight to 10 years after FDA approval. A qualitative study [39] reported that many experts were worried about this trends, and a participating national expert remarked: "*The incentive to do really properly conducted trials—after the device is already approved? It's so low.*"".

## PMA supplement and amendment pathway

Studies [40–42] reported that manufactures increasingly made minor changes to dermatologic devices and got them approved through the PMA supplement and amendment pathway without any clinical data. The studies found an association between PMA supplements and an 30% increased risk of any recall, and that for the FDA approval application of high-risk device modifications, fewer than half were randomized.

## The 510k FDA pathway

It was reported [43] that there was limited evidence on efficacy and harms of a random sample of 1000 medical devices cleared through the 510(k) pathway, and further reported that only 17.5% of the 510(k) approved devices had published research, and that one fourth of studies reported conflicts of interests. Also, it was observed that 4.3% of FDA 510(k) approved medical devices was approved based on a predicate medical device with an ongoing recall, and an association between the use of a predicate medial device with a history of or an ongoing recall for an FDA 510(k) clearanceand a

subsequent recall was observed [44,45]. Finally, data suggests that (the PMA pathway is more costly and time-consuming than the 510(k) pathway, and it was argued by the study authors that strict regulatory requirements could limit device innovation [46].

### Predicate devices

It has been documented [47,48] that vaginal meshes and surgical meshes approved through the FDA 510(k) pathway are connected in an ancestral network of equivalence claims, and it wasobserved that scientific evidence on substantial equivalence was not publicly available, despite a legal requirement [49].

### The European Union

A lack of public data within the EU was noted [15,50], and an e-mail was shown[15] from a private notified body reply: "*[]… The notifying body is a client working on behalf of the manufacturer and sees the clinical data as being commercially sensitive.*". A German study [51] from 2019 examined applications from a large German ethics committee which oversee drug and medical device studies conducted in the federal state of Berlin, and reported that 57% of applications planned to use a randomised controlled trial for their premarket study. Many medical devices are introduced to the European before the American market [52], however, medical devices approved in the EU before in the USA are associated with an increased risk of postmarketing safety alerts and recalls. A recent study [53] from 2023 concluded that conformité européenne certification for a medical device in Europe does not guarantee efficacy, but only that the device is compliant to the EU law, and that lack of transparency of data hampers evidence-based decision-making.

### Implementation

A total of 17 studies reported on implementation, in the following section eight studies will be mentioned. Clinical evidence was the most important factor in the choice of new protheses according to 90 orthopaedic surgeons [54]. In a qualitative study [55] from 2016 a conflict between experience and evidence-based decision-making was observed. A wound care nurse was also reported to have said: "*They're relying mostly on your expertise, so you don't need to make, like, a huge argument. You just have to say, 'Within my experience on a daily basis, I find that these do not work for patients.*". Also, it was reported [56] that hospital implementation decisions are increasingly likely to be based on total cost including volume discount. In a qualitative study [57] from 2013 a head of a Hospital department said: "*I have a selection of [devices] sitting in front of me on my desk which I would be hesitant to use with a horse, but they are what the market provides because everything has to be as cheap as possible*".

The clinical evidence submitted by manufacturers to a decision committee was examined, and the authors concluded that the limited clinical evidence which hospital decision-makers rely on poses a big problem [58]. In a qualitative study [59] from 2017 an interviewee from a European health technology assessment institutions also expressed concerns regarding lack of evidence: "*[...] Very often it is so, that if somebody asks us about a new device, there is hardly any published studies, which is usually not the case for drugs.*". Furthermore, a qualitative study [60] from 2017 also reported that missing evidence on device performance was an issue, and an orthopaedic specialist said: "*There is really nothing in the literature that is helpful on this. Everything we do in orthopedics is a beta test. You know how drugs go through phases of testing? There's nothing equivalent to that in orthopedics. … []*".

According to a Canadian qualitative study [61] that explored interactions between hospital employees and the device industry concluded that further research is warranted to establish the clinical implications of these interactions. The study reported that most cardiovascular or orthopaedic surgeons described symbiotic relationships with the device industry representatives, and one participant remarked that: "*we can't get along without each other*". Furthermore, interviewees

expressed that the representatives often were in the operating room. Two interviewees remarked that: "*My rep is there 95% of the time or more*" and "*They would come to all the implantations*".

## Postmarket surveillance

A total of 40 studies reported on postmarket surveillance, in the following section 12 studies will be mentioned. The MAUDE database was used to compare complication profiles of urogynaecology meshes,[62], however, the database was criticized for limited postmarket adverse event reporting [63], and inaccuracies in the verbatim description of each report [64,65]. Physicians of varying specialties were reported [66] to find adverse event reporting for medical devices futile, not possible or unnecessary. Studies [13,67–70] that compared the recall rates of the PMA and 510(k) pathway reported inconsistent results, where some found that 510(k) had higher risk of recalls than the PMA pathway and vice-versa. Studies indicate that electronic hospital clinical data as a data source could be used to assess the efficacy and harms of medical devices in use [71,72].

In summary, we identified a lot of studies investigating medical devices. Studies conclude that the evidence on medical devices at approval is not sufficient, and there is a trend towards a less strict demand of evidence in the FDA over time. Very little evidence on medical device approval in the EU is available because a lack of publicly available data. Implementation of medical devices into clinical practice is difficult because evidence of the medical devices is lacking. Pricing and clinical expertise seems to be the drivers of implementation, however, research on the decision-making process is also lacking. Additionally, the close relationship between clinicians and medical device manufacturers requires more thorough examination because conflicts of interests might influence decision-making. Postmarket surveillance studies were mostly conducted using FDA databases. The databases do not contain high quality data, and many studies suggests the use of registry-based studies based on countries' own data collection. To provide data for such studies, it would require the creation of high-quality public registries with unique device identifiers using electronic hospital records.

## Discussion

We included 139 primary studies on medical device regulation, implementation and postmarket surveillance. Studies revealed that it was difficult for regulatory agencies and health care providers to know what clinical evidence suffice for regulatory approval and subsequent implementation of medical devices, however, for various medical devices the body of evidence was considered inadequate. Most studies used the FDA as their primary data source, and only few examined medical devices within EU.

The lack of evidence on medical devices observed in the included studies of our review is in accordance with a systematic review [73] of peer-reviewed papers on high-risk orthopaedic devices. The review reported that peer-reviewed literature alone is an insufficient source of clinical information on efficacy and harms of medical devices, and the authors recommended to add registry data to the body of evidence. Also, a Canadian scoping review [74] concluded that specific criteria for decision-making in the early adoption phase of novel medical devices used for surgery are lacking. Additionally, we identified limited research on how and why specific medical devices were selected and adopted into clinical practice. The need for more empirical work on factors involved in implementation is in agreement with a recent systematic review [75]. Problems with the databases used for research within medical devices have also been highlighted in a recent opinion piece [76], in which the authors requested improved reporting to the database. The authors stated that 97% of the reports in the MAUDE database were submitted to the FDA by the device manufacturers [77], and recommended increased reporting to the database directly by physicians and patients. Furthermore, a systematic review [78] reported that the quality of cardiovascular surgery studies that used the MAUDE database were considered poor because of reasons such as lack of consecutive patients and undetermined follow-up. The European project Coordinating Research and Evidence for Medical Devices conducted a systematic review and found that data quality of the European registries needs improvements [79].

As mentioned, concerns about approval and surveillance of medical devices have already been raised [14,15]. We acknowledge that certain devices may still be implemented despite limited data due to factors such as urgency or a lack of alternative treatment options. Additionally, conducting high-quality studies in some areas can be challenging. For instance, when orthopaedic implants are used as part of an inconsistent intervention where several covariables may be associated with clinical outcomes, e.g., that fractures are heterogeneous, and that surgical procedures and experience varies between surgeons and hospitals. Also, methodological factors such as blinding are difficult to do in surgical studies. Furthermore, when the field range from scalpels to pacemakers it can be hard to know what evidence suffice, and it is often very case and context specific. Regardless of the complexities the approval of medical devices should be improved to ensure patient safety and public trust. The idea, development, exploration, assessment, long-term follow-up of device innovation (IDEAL-D) [80] framework was created in 2016 to address the lack of evidence in the medical device field by providing guidance on how to properly evaluate medical devices at each stage of the development by suggesting study designs to for instance demonstrate efficacy and monitor harms. However, the framework has not been widely implemented, possibly due to limited understanding of the IDEAL recommendations and how to apply them [81]. Regulatory agencies have a responsibility to ensure patient safety, even when the task is difficult. As such, they should demand that medical devices are properly investigated before they are introduced to the market. However, improving evidence standards also presents challenges, such as potential delays in access to interventions, increased resource demands for regulatory bodies, and the challenge of ensuring that rigorous evidence requirements remain practically feasible.

Clinical researchers need to push for open non-industry influenced registries with high-quality data with both efficacy data and adverse reports. These could be valuable adjacent to explore long-terms harms of medical devices. Especially since the regulatory pathways are not likely – anytime soon – to improve. On the contrary, a tendency to less strict evidence requirement at approval, and that mandated postapproval studies frequently seems to be of poorer methodological quality and remain unpublished was observed in the included studies. Although, the FDA should be praised for being more transparent in sharing data compared to current system in EU, the quality of the data within the MAUDE database should be strengthened. The FDA has introduced unique device identifiers for a long time, however, implementation is still lacking behind [82]. The initiative The National Evaluation System for health Technology Coordinating Center [83] was created for the purpose of strengthening medical device postmarket surveillance in the USA. In EU unique device identifiers are first now mandated and implemented through the EU medical device regulation from 2017. EUDAMED [17] should be used to mitigate the scarcity of research on medical devices within the EU, when it is fully implemented. As the full implementation of the regulation and EUDAMED [17] has been delayed, results might first be available within a few years, and it is not yet clear the exact amount of data that will be publicly available.

Conflicts of interests are also a concern because of the close relationship between manufacturers and clinical practice. Decision-makers who implement medical devices should be cautious hereof. The extent and consequences of this relationship remains to be elucidated.

Many medical devices lack clinical efficacy studies. Additionally, postmarket surveillance studies, which monitor these devices, are hindered by poor-quality data in publicly available databases. As a result, accurately assessing patient benefits and harms remains challenging. Robust evidence is essential for decision-makers to allocate resources effectively, ensuring that high-impact medical devices are prioritized over those with minimal benefit—especially in healthcare systems facing financial constraints. While stricter regulations may increase costs and potentially slow innovation, patient safety must take precedence. Unlike cars, cardiovascular implants and similar medical devices cannot be easily recalled, emphasizing the need for thorough premarket evaluation and reliable postmarket surveillance. Above all, patients deserve the confidence that the medical devices used in their treatment are both effective and safe.

 

## Limitations

The study was limited by a pragmatical specific search strategy by using broader medical device keywords potentially missing relevant records, and by excluding in vitro diagnostics and software as a medical device not used for treatment purposes. The increasing use of artificial intelligence in diagnostics, prediction models and software as a medical device will call for regulatory authorities and researchers to evaluate and improve the regulatory framework for this group of medical devices. Since most studies relied on FDA databases and only a few studies were conducted with European data, results may not be generalizable to Europe or other non-USA countries. Also, the potential lack of data integrity in the FDA databases, due to possible inaccuracies, missing data, and underreporting, may have influenced the results of the included studies. The aim of the scoping review was to collect studies that examined a broad range of medical devices, and the reported inadequacy of evidence may not apply to all types of medical devices. Finally, as with any review, selection bias may have occurred if relevant studies were not identified or remained unpublished due to their findings.

## Conclusion

Studies on medical devices are mainly conducted using FDA device databases, since restricted access to publicly available data has hindered research within the EU. Analyses from studies of regulatory approval applications submitted to the FDA have generally found the evidence supporting medical devices to be of low quality. The lack of robust evidence complicates selection and adoption of medical devices, and there is limited research on how and why a specific medical device is chosen over another. We suggest that evidence on medical device efficacy and harms should be strengthened through higher demands on well-conducted randomised and non-randomised observational studies from regulatory agencies, improvement of accessible registries for postmarket surveillance to monitor harms, and enhancement of the quantity and quality of data within these registries. Additionally, international collaboration in medical device regulation and surveillance could facilitate harmonization between FDA and EU regulatory practices, improving data consistency and overall patient safety.

## Supporting information

**S1 Table. Population, Concepts and Context.**
(PDF)

**S1 File. Supporting information.**
(PDF)

**S2 File. Excluded studies.**
(DOCX)

## Author contributions

**Conceptualization:** Mathias Damkjær, Asbjørn Hróbjartsson, Jeppe B. Schroll.

**Data curation:** Mathias Damkjær, Mia Elkjær, Jeppe B. Schroll.

**Formal analysis:** Mathias Damkjær.

**Methodology:** Mathias Damkjær, Asbjørn Hróbjartsson, Jeppe B. Schroll.

**Project administration:** Mathias Damkjær, Jeppe B. Schroll.

**Supervision:** Asbjørn Hróbjartsson, Jeppe B. Schroll.

**Visualization:** Mathias Damkjær.

**Writing – original draft:** Mathias Damkjær, Mia Elkjær, Asbjørn Hróbjartsson, Jeppe B. Schroll.

**Writing – review & editing:** Mathias Damkjær, Mia Elkjær, Asbjørn Hróbjartsson, Jeppe B. Schroll.

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
