## [Decision Letter · Decision Letter 0]

7 Feb 2025

PONE-D-24-48501Scoping review on regulation, implementation and postmarket surveillance of medical devicesPLOS ONE

Dear Dr. Damkjær,

Thank you for submitting your manuscript to PLOS ONE. After careful consideration, we feel that it has merit but does not fully meet PLOS ONE’s publication criteria as it currently stands. Therefore, we invite you to submit a revised version of the manuscript that addresses the points raised during the review process.

Kindly address the comments and clarifications requested by the reviewers below. 

We look forward to receiving your revised manuscript.

Kind regards,

Swarnali Goswami

Academic Editor

PLOS ONE

Journal Requirements:

2.  Please include captions for your Supporting Information files at the end of your manuscript, and update any in-text citations to match accordingly. Please see our Supporting Information guidelines for more information: http://journals.plos.org/plosone/s/supporting-information .

Reviewers' comments:

Reviewer's Responses to Questions

**Comments to the Author**

1. Is the manuscript technically sound, and do the data support the conclusions?

Reviewer #1: Partly

Reviewer #2: Yes

2. Has the statistical analysis been performed appropriately and rigorously? 

Reviewer #1: No

Reviewer #2: I Don't Know

3. Have the authors made all data underlying the findings in their manuscript fully available?

Reviewer #1: Yes

Reviewer #2: Yes

4. Is the manuscript presented in an intelligible fashion and written in standard English?

Reviewer #1: No

Reviewer #2: No

5. Review Comments to the Author

Reviewer #1: The manuscript provides valuable insights into the current state of medical device regulation, highlighting the gaps in evidence quality and the challenges faced by regulatory agencies and healthcare providers. While the study is well-organized, several areas require further clarification and detail. I hope you find my comments helpful. Some specific items for clarification are outlined below:

• Introduction: The opening sentence could be strengthened by briefly mentioning the challenges faced in medical device regulation (e.g., variability in approval standards, post market surveillance issues) before delving into statistics or regulatory processes.

• Introduction: the explanation of the FDA’s approval process can be improved. Focus on the core differences between the more rigorous PMA pathway and the 510(k) process.

• Introduction: Explicitly highlight the research gap this review is trying to fill

• Methods: The rationale for excluding secondary papers is clear, but the use of "secondary papers" might be confusing. It could be useful to explicitly define this term or rephrase it to avoid ambiguity.

• Methods: The mention of the review transitioning between being classified as a "scoping" or "mapping" review is interesting but could be simplified to avoid unnecessary detail in this section. Condense it to: "We ultimately settled on a scoping review due to the nature of the studies included."

• Methods: Instead of using both "categorizing" and "mapping," stick to one term for consistency.

• Results: The process of identifying and including studies is clearly described, but the phrasing in some places can be simplified. consider: "The search identified 4,904 records, of which 3,862 titles/abstracts were screened after de-duplication."

• Results: The section on post market surveillance contains several useful details but could be more concise. Some sentences are a bit repetitive, especially when describing similar databases or trends.

• Results: Some phrases could be streamlined to avoid redundancy. For example, instead of repeating "two studies" or "three studies," consider summarizing the findings from these studies collectively.

• Discussion: When introducing the IDEAL-D framework, it's important to explain its purpose and why it hasn't been widely implemented.

• Discussion: While evidence gaps are a concern, the section does not explore the nuances of why certain devices might still be implemented without enough data (e.g., urgency, limited alternatives, etc.).

• Discussion: While the discussion mentions that regulatory agencies should improve their evidence requirements, it doesn’t explore the potential consequences of these improvements in depth. What are the practical, ethical, and logistical challenges of improving evidence standards? These could be explored further.

• Limitations: Can be expanded.

o There could be a potential selection bias in the included studies, as many of them relied on FDA databases, and only a few included studies from the EU or other regions. This geographical and methodological limitation may affect the generalizability of the results.

o The quality of the data used in the included studies varied, particularly in relation to database-based research. Issues such as missing data, underreporting, and potential inaccuracies in the FDA’s MAUDE database could have impacted the findings.

o Medical devices vary widely in their complexity, use, and risk profiles (from simple bandages to life-saving pacemakers). This heterogeneity could have led to difficulties in generalizing the findings across different types of devices and in assessing the adequacy of evidence for such a diverse range.

o There may be publication bias in the included studies, as studies with negative or inconclusive findings might be less likely to be published or included in databases. This could result in an overrepresentation of studies with positive or supportive findings on regulatory practices.

o There may be publication bias in the included studies, as studies with negative or inconclusive findings might be less likely to be published or included in databases. This could result in an overrepresentation of studies with positive or supportive findings on regulatory practices.

• Conclusion: Consider stressing the importance of international collaboration in medical device regulation and surveillance. Since the regulatory practices in the FDA and EU differ, there may be opportunities for harmonization to improve both systems.

• Conclusion: The recommendation to "strengthen evidence" is useful, but it could be more specific. For example, what types of evidence are most critical (e.g., long-term efficacy, post-market surveillance data)? Also, you could discuss how these improvements would directly impact patient safety, healthcare costs, and device innovation.

Reviewer #2: Pg 4 Paragraph 2 line 2: The word “notifying” should be “notified” as they are called Notified Bodies.

“the FDA and private notifying notified bodies within Europe. Private notifying notified bodies are organisations or”

Please make the above correction throughout the document.

Pg 4 Paragraph 2 line 3: “EU Member State” or “EU Member Country” might be a better phrase instead of “EU Country” in my opinion. Please update throughout the document as applicable.

Pg 4 Paragraph 2 line 4: I suggest adding “applicable” to the sentence. Just because a device has a CE mark doesn’t mean it meets all EU regulations. Example -

“devices have met applicable European safety, health and environmental protection requirements”

Pg 4 paragraph 2 Line 5: “devices are classified by both the FDA and the EU into three risk classes I-III.”

I assume you are trying to summarize the classes here for the point you are making in the next sentence and for ease of data charting but I would suggest clarifying that classification in the EU is Class I, IIa,IIb, and III. You may also add a sentence after this such as “Moreover, in EU Class II is further subdivided into Class IIa and IIb” just to be accurate as they are important subclassifications. Class I is also subdivided for that matter. This is not a dealbreaker but just a recommendation for accuracy. You may choose to keep it just Class II for data charting and should indicate that in the relevant section of the manuscript.

Pg 4 Paragraph 3 line 6: ”substantial equivalence to a previous approved device (after 1976),”

Whenever you talk about 510k submissions, always say “cleared” and not “approved”. PMA applications are “approved” by the FDA whereas 510k application are “cleared” by the FDA and they are called “clearances” not “approvals”. Please correct where applicable throughout the document.

Pg 4 Paragraph 3 line 13: “Among high-risk devices approved between 2008 and 2017 through FDA PMA approval, 27% were recalled at nine years”.

According to the paper cited (citation# 13), the study period was January 1, 2008, and December 31, 2017, which is actually 10 years if you are using that time period. Moreover, the recall data was actually collected till 2019 which is 12 years. However, in the paper, I also found that statement under discussion which says “At 9 years after device approval, the risk of any recall for devices with PMA and 510(k) clearance is 32% vs 13%”. But, there is also a statement in the abstract that states “During the study period, 28 246 devices received 510(k) clearance and 310 devices (10.7%) received PMA; 3012 devices (10.7%) with 510(k) clearance and 84 devices (27.1%) with PMA were recalled.” and I assume that is where the 27% in your statement comes from. Please re-confirm and clarify this statement, if necessary.

Pg 5 Paragraph 1 Line 1-2: “Regulatory institutions such as the EU promised more stringent and transparent medical device regulation with an EU directive from 2017 (16).”

Technically, “European Commission” is the regulatory institution in the EU and not the European Union itself. I would suggest a correction for accuracy. Secondly, there is a difference between a Directive and a Regulation. The 2017 regulations (EU MDR and EU IVDR) are not directives. Please correct the statement accordingly. You may also see this link for more information - https://european-union.europa.eu/institutions-law-budget/law/types-legislation_en

Pg 5 Paragraph 2 Line 3: “Studies that examine medical devices are thought to be very heterogeneous, that is why a scoping review as study design is appropriate to create an overview.’

I think there is a typo. See red text above and make the correction.

Pg 6 Paragraph 2 Line 1-2: “software assisted medical devices, e.g., artificial intelligence solutions”

I think you mean “Software as a Medical Device” (SaMD), which is the technically correct name used universally across jurisdictions.. I suggest you correct it.

Pg 8 Paragraph 6 Line 1: “A totalt of 68 studies reported on medical devices in relation to approval.”

Correct the typographical error in the word “total”.

Pg 9 Paragraph 1 Line 1: “We identified 12 studies that all all concluded that premarket approval studies did not enroll”. The word “all” is repeated. Please correct.

Pg 9 Paragraph 1 Line 2: “nonwhite” should have a hyphen in between i.e., non-white.

Pg 10 Paragraph 2 Line 11: “not guarantee efficacy, but only that the device is complient to the EU law, and that lack of”

Correct the spelling of the word “Compliant”.

6. PLOS authors have the option to publish the peer review history of their article (what does this mean? ). If published, this will include your full peer review and any attached files.

**Do you want your identity to be public for this peer review?** For information about this choice, including consent withdrawal, please see our Privacy Policy .

Reviewer #1: No

Reviewer #2: No

---

## [Author Response · Author response to Decision Letter 1]

14 Apr 2025

Manuscript PONE-D-24-48501 Copenhagen 22nd of March 2025

Dear Swarnali Goswami,

Thank you for giving us the opportunity to submit a revised draft of our manuscript: "Scoping re-view on regulation, implementation and postmarket surveillance of medical devices" to PLOS One. We appreciate all the time you and peer-reviewers have put into providing us with detailed feedback on our manuscript.

We have incorporated the suggestions made by the reviewers and have attached both a clean and tracked version of the manuscript.

On behalf of all authors,

Mathias Weis Damkjær

Reviewer #1:

The manuscript provides valuable insights into the current state of medical device regulation, highlighting the gaps in evidence quality and the challenges faced by regulatory agencies and healthcare providers. While the study is well-organized, several areas require further clarification and detail. I hope you find my comments helpful. Some specific items for clarification are outlined below:

Authors reply: Thank you.

• Introduction: The opening sentence could be strengthened by briefly mentioning the challenges faced in medical device regulation (e.g., variability in approval standards, post market surveillance issues) before delving into statistics or regulatory processes.

Authors reply: We have added the following sentence: "Medical devices are fundamental to medi-cine, and the basis for a growing billion-dollar industry (1,2). The World Health Organization es-timates that there are more than two million medical devices on the world market and they range from pacemakers, artificial intelligence software to simple instruments such as scalpels (3). The regulation of medical devices faces several challenges, including inconsistent approval stand-ards across different devices and countries, as well as difficulties in monitoring device harms and efficacy once they are in use. ". In the introduction section.

• Introduction: the explanation of the FDA’s approval process can be improved. Focus on the core differences between the more rigorous PMA pathway and the 510(k) process.

Authors reply: We have now added the core differences between PMA approval and 510(k) clear-ance more clearly in the section in the introduction describing these.

• Introduction: Explicitly highlight the research gap this review is trying to fill

Authors reply: We have revised the aim of our introduction to: "We did not identify prior reviews that created an overview of the conducted research on medical devices across the selected key lifecycle domains - approval, implementation and postmarket surveillance - likely because the studies are very heterogeneous. Additionally, we hypothesized that studies investigating why one medical device is adopted over a similar alternative are limited. Our aim was to systematically map and characterize empirical studies on regulation, postmarket surveillance and implementation of medical devices in a scoping review given the heterogeneity of studies. We sought to discuss strengths and limitations of these domains, based on a condensation of the predominant conclu-sions from included studies, while also identifying areas that could be improved, and where more research is needed." Hopefully this will make it more clear why we think a scoping review in the area is relevant.

• Methods: The rationale for excluding secondary papers is clear, but the use of "secondary papers" might be confusing. It could be useful to explicitly define this term or rephrase it to avoid ambigui-ty.

Authors reply: We have revised it to: "We excluded secondary papers, defined as publications that do not present primary empirical research, such as editorials, systematic reviews based on biblio-graphic databases, editorial letters, and opinion papers. Additionally, we excluded papers that did not fit the predefined PCC framework."

• Methods: The mention of the review transitioning between being classified as a "scoping" or "mapping" review is interesting but could be simplified to avoid unnecessary detail in this section. Condense it to: "We ultimately settled on a scoping review due to the nature of the studies includ-ed."

Authors reply: We have chosen to refer to the protocol amendments and delete this part as we agree that it is an unnecessary detail.

• Methods: Instead of using both "categorizing" and "mapping," stick to one term for consistency.

Authors reply: We have chosen "mapping/mapped" and corrected throughout the manuscript.

• Results: The process of identifying and including studies is clearly described, but the phrasing in some places can be simplified. consider: "The search identified 4,904 records, of which 3,862 ti-tles/abstracts were screened after de-duplication."

Authors reply: We have simplified the phrasing to: "The search identified 4,904 records. After de-duplication, 3,862 title/abstracts were screened and 368 full-text studies were assessed, out of which 141 met eligibility criteria, and 139 were included in the final analysis"

• Results: The section on post market surveillance contains several useful details but could be more concise. Some sentences are a bit repetitive, especially when describing similar databases or trends.

Authors reply: We have now revised the sentences describing the studies based on the MAUDE database, as we believe these were the most repetitive. We have written: "The MAUDE database was used to compare complication profiles of urogynaecology meshes (62), however, the database was criticized for limited postmarket adverse event reporting (63), and inaccuracies in the verba-tim description of each report (64,65)."

• Results: Some phrases could be streamlined to avoid redundancy. For example, instead of repeat-ing "two studies" or "three studies," consider summarizing the findings from these studies collec-tively.

Authors reply: We have now revised the result section and tried to write it more concise and with active language, and we have avoided mentioning the number of studies. As the changes are nu-merous, we here refer to the tracked manuscript document.

• Discussion: When introducing the IDEAL-D framework, it's important to explain its purpose and why it hasn't been widely implemented.

Authors reply: We have now clarified this in the discussion: "The idea, development, exploration, assessment, long-term follow-up of device innovation (IDEAL-D) (81) framework was created in 2016 to address the lack of evidence in the medical device field by providing guidance on how to properly evaluate medical devices at each stage of the development by suggesting study designs to for instance demonstrate efficacy and monitor harms. However, the framework has not been wide-ly implemented, possibly due to limited understanding of the IDEAL recommendations and how to apply them (82)."

• Discussion: While evidence gaps are a concern, the section does not explore the nuances of why certain devices might still be implemented without enough data (e.g., urgency, limited alterna-tives, etc.).

Authors reply: We have now revised it to the following: "We acknowledge that certain devices may still be implemented despite limited data due to factors such as urgency or a lack of alterna-tive treatment options. Additionally, conducting high-quality studies in some areas can be chal-lenging."

• Discussion: While the discussion mentions that regulatory agencies should improve their evi-dence requirements, it doesn’t explore the potential consequences of these improvements in depth. What are the practical, ethical, and logistical challenges of improving evidence standards? These could be explored further.

Authors reply: We agree and have now allude to these important issues in the discussion "Howev-er, improving evidence standards also presents challenges, such as potential delays in access to interventions, increased resource demands for regulatory bodies, and the challenge of ensuring that rigorous evidence requirements remain practically feasible."

• Limitations: Can be expanded.

Authors reply: We have now added all the points below to the limitations section: "Since most studies relied on FDA databases and only a few studies were conducted with European data, results may not be generalizable to Europe or other non-USA countries. Also, the potential lack of data integrity in the FDA databases, due to possible inaccuracies, missing data, and underreporting, may have influenced the results of the included studies. The aim of the scoping review was to col-lect studies that examined a broad range of medical devices, and the reported inadequacy of evi-dence may not apply to all types of medical devices. Finally, as with any review, selection bias may have occurred if relevant studies were not identified or remained unpublished due to their findings."

o There could be a potential selection bias in the included studies, as many of them relied on FDA databases, and only a few included studies from the EU or other regions. This geographical and methodological limitation may affect the generalizability of the results.

Authors reply: This point has been added to the limitations.

o The quality of the data used in the included studies varied, particularly in relation to database-based research. Issues such as missing data, underreporting, and potential inaccuracies in the FDA’s MAUDE database could have impacted the findings.

Authors reply: This point has been added to the limitations.

o Medical devices vary widely in their complexity, use, and risk profiles (from simple bandages to life-saving pacemakers). This heterogeneity could have led to difficulties in generalizing the find-ings across different types of devices and in assessing the adequacy of evidence for such a diverse range.

Authors reply: This point has been added to the limitations.

o There may be publication bias in the included studies, as studies with negative or inconclusive findings might be less likely to be published or included in databases. This could result in an overrepresentation of studies with positive or supportive findings on regulatory practices.

Authors reply: This point has been added to the limitations.

o There may be publication bias in the included studies, as studies with negative or inconclusive findings might be less likely to be published or included in databases. This could result in an overrepresentation of studies with positive or supportive findings on regulatory practices.

Authors reply: Phrase duplication. Addressed above.

• Conclusion: Consider stressing the importance of international collaboration in medical device regulation and surveillance. Since the regulatory practices in the FDA and EU differ, there may be opportunities for harmonization to improve both systems.

Authors reply: This is a great point, and we have added this to the conclusion: "Additionally, inter-national collaboration in medical device regulation and surveillance could facilitate harmonization between FDA and EU regulatory practices, improving data consistency and overall patient safety."

• Conclusion: The recommendation to "strengthen evidence" is useful, but it could be more specif-ic. For example, what types of evidence are most critical (e.g., long-term efficacy, post-market surveillance data)? Also, you could discuss how these improvements would directly impact patient safety, healthcare costs, and device innovation.

Authors reply: We have now added suggestions for the types of evidence in the conclusion. Fur-thermore, we have revised the discussion to incorporate why the improvements are important: "Many medical devices lack clinical efficacy studies. Additionally, postmarket surveillance stud-ies, which monitor these devices, are hindered by poor-quality data in publicly available databases. As a result, accurately assessing patient benefits and harms remains challenging. Robust evidence is essential for decision-makers to allocate resources effectively, ensuring that high-impact medi-cal devices are prioritized over those with minimal benefit—especially in healthcare systems fac-ing financial constraints. While stricter regulations may increase costs and potentially slow inno-vation, patient safety must take precedence. Unlike cars, cardiovascular implants and similar med-ical devices cannot be easily recalled, emphasizing the need for thorough premarket evaluation and reliable postmarket surveillance. Above all, patients deserve the confidence that the medical devices used in their treatment are both effective and safe."

Reviewer #2:

Authors reply: First and foremost, thank you kindly for your detailed review. We have incorpo-rated all your good comments into the manuscript.

Pg 4 Paragraph 2 line 2: The word “notifying” should be “notified” as they are called Notified Bodies.

“the FDA and private notifying notified bodies within Europe. Private notifying notified bodies are organisations or”

Please make the above correction throughout the document.

Authors reply: We have corrected this throughout the document.

Pg 4 Paragraph 2 line 3: “EU Member State” or “EU Member Country” might be a better phrase instead of “EU Country” in my opinion. Please update throughout the document as applicable.

Authors reply: We have edited this to: "EU Member Country".

Pg 4 Paragraph 2 line 4: I suggest adding “applicable” to the sentence. Just because a device has a CE mark doesn’t mean it meets all EU regulations. Example -

“devices have met applicable European safety, health and environmental protection requirements”

Authors reply: "applicable" has now been added to the sentence.

Pg 4 paragraph 2 Line 5: “devices are classified by both the FDA and the EU into three risk classes I-III.”

I assume you are trying to summarize the classes here for the point you are making in the next sen-tence and for ease of data charting but I would suggest clarifying that classification in the EU is Class I, IIa,IIb, and III. You may also add a sentence after this such as “Moreover, in EU Class II is further subdivided into Class IIa and IIb” just to be accurate as they are important subclassifica-tions. Class I is also subdivided for that matter. This is not a dealbreaker but just a recommenda-tion for accuracy. You may choose to keep it just Class II for data charting and should indicate that in the relevant section of the manuscript.

Authors reply: You are right, and it is a good point. We have added: "Medical devices are classi-fied by both the FDA and the EU into three risk classes I-III, which are further subdivided (6,7)." FDA also have subdivision in both class I and II (with and without exemptions).

Pg 4 Paragraph 3 line 6: ”substantial equivalence to a previous approved device (after 1976),”

Whenever you talk about 510k submissions, always say “cleared” and not “approved”. PMA appli-cations are “approved” by the FDA whereas 510k application are “cleared” by the FDA and they are called “clearances” not “approvals”. Please correct where applicable throughout the document.

Authors reply: Thank you. This has now been corrected throughout the document.

Pg 4 Paragraph 3 line 13: “Among high-risk devices approved between 2008 and 2017 through FDA PMA approval, 27% were recalled at nine years”.

According to the paper cited (citation# 13), the study period was January 1, 2008, and December 31, 2017, which is actually 10 years if you are using that time period. Moreover, the recall data was actually collected till 2019 which is 12 years. However, in the paper, I also found that state-ment under discussion which says “At 9 years after device approval, the risk of any recall for de-vices with PMA and 510(k) clearance is 32% vs 13%”. But, there is also a statement in the abstract that states “During the study period, 28 246 devices received 510(k) clearance and 310 devices (10.7%) received PMA; 3012 devices (10.7%) with 510(k) clearance and 84 devices (27.1%) with PMA were recalled.” and I assume that is where the 27% in your statement comes from. Please re-confirm and clarify this statement, if necessary.

Authors reply: Thank you so much for noticing this. We have revised it to: "as of December 31, 2019" instead of "… at nine years".

The other probability percentages are from the Cox regression that adjust for diff

---

## [Decision Letter · Decision Letter 1]

12 May 2025

Scoping review on regulation, implementation and postmarket surveillance of medical devices

PONE-D-24-48501R1

Dear Dr. Damkjær,

We’re pleased to inform you that your manuscript has been judged scientifically suitable for publication and will be formally accepted for publication once it meets all outstanding technical requirements.

Kind regards,

Swarnali Goswami

Academic Editor

PLOS ONE

Additional Editor Comments (optional):

Reviewers' comments:

Reviewer's Responses to Questions

**Comments to the Author**

1. If the authors have adequately addressed your comments raised in a previous round of review and you feel that this manuscript is now acceptable for publication, you may indicate that here to bypass the “Comments to the Author” section, enter your conflict of interest statement in the “Confidential to Editor” section, and submit your "Accept" recommendation.

Reviewer #2: All comments have been addressed

2. Is the manuscript technically sound, and do the data support the conclusions?

Reviewer #2: Yes

3. Has the statistical analysis been performed appropriately and rigorously? 

Reviewer #2: I Don't Know

4. Have the authors made all data underlying the findings in their manuscript fully available?

Reviewer #2: Yes

5. Is the manuscript presented in an intelligible fashion and written in standard English?

Reviewer #2: Yes

6. Review Comments to the Author

Reviewer #2: Thank you for addressing all the comments and recommendations in your revision. The manuscript offers important perspectives on the present landscape of medical device regulation, emphasizing deficiencies in evidence quality and the difficulties encountered by regulatory agencies and the medical device industry.

7. PLOS authors have the option to publish the peer review history of their article (what does this mean? ). If published, this will include your full peer review and any attached files.

**Do you want your identity to be public for this peer review?** For information about this choice, including consent withdrawal, please see our Privacy Policy .

Reviewer #2: No

---

## [Editor Report · Acceptance letter]

PONE-D-24-48501R1

PLOS ONE

Dear Dr. Damkjær,

I'm pleased to inform you that your manuscript has been deemed suitable for publication in PLOS ONE. Congratulations! Your manuscript is now being handed over to our production team.

Kind regards,

on behalf of

Dr. Swarnali Goswami

Academic Editor

PLOS ONE